# Innovative Low-Cost Composite Nanoadsorbents Based on Eggshell Waste for Nickel Removal from Aqueous Media

**DOI:** 10.3390/nano13182572

**Published:** 2023-09-16

**Authors:** Adina-Elena Segneanu, Roxana Trusca, Claudiu Cepan, Maria Mihailescu, Cornelia Muntean, Dumitru Daniel Herea, Ioan Grozescu, Athanasios Salifoglou

**Affiliations:** 1Institute for Advanced Environmental Research, West University of Timisoara (ICAM-WUT), 4 Oituz St., 300086 Timișoara, Romania; adina.segneanu@e-uvt.ro; 2National Center for Micro and Nanomaterials, Politehnica University of Bucharest, Str. Splaiul Independenţei, Nr. 313, 060042 Bucharest, Romania; truscaroxana@yahoo.com; 3Department of Applied Chemistry and Engineering of Inorganic Compounds and the Environment, University Politehnica Timisoara, Piata Victoriei Nr. 2, 300006 Timisoara, Romania; cepanclaudiu@gmail.com (C.C.); mihailescumia@gmail.com (M.M.); cornelia.muntean@upt.ro (C.M.); ioangrozescu@gmail.com (I.G.); 4Research Institute for Renewable Energy, 138 Gavril Musicescu St., 300501 Timisoara, Romania; 5National Institute of Research and Development for Technical Physics, 47 Mangeron Blvd, 700050 Iasi, Romania; dherea@phys-iasi.ro; 6Laboratory of Inorganic Chemistry and Advanced Materials, School of Chemical Engineering, Aristotle University of Thessaloniki, 54124 Thessaloniki, Greece

**Keywords:** composite nanoadsorbent, waste reuse, sustainable economy, wastewater remediation, heavy metals, eggshell, zeolite, nickel adsorption

## Abstract

In a contemporary sustainable economy, innovation is a prerequisite to recycling waste into new efficient materials designed to minimize pollution and conserve non-renewable natural resources. Using an innovative approach to remediating metal-polluted water, in this study, eggshell waste was used to prepare two new low-cost nanoadsorbents for the retrieval of nickel from aqueous solutions. Scanning electron microscopy (SEM) results show that in the first eggshell–zeolite (EZ) adsorbent, the zeolite nanoparticles were loaded in the eggshell pores. The preparation for the second (iron(III) oxide-hydroxide)–eggshell–zeolite (FEZ) nanoadsorbent led to double functionalization of the eggshell base with the zeolite nanoparticles, upon simultaneous loading of the pores of the eggshell and zeolite surface with FeOOH particles. Structural modification of the eggshell led to a significant increase in the specific surface, as confirmed using BET analysis. These features enabled the composite EZ and FEZ to remove nickel from aqueous solutions with high performance and adsorption capacities of 321.1 mg/g and 287.9 mg/g, respectively. The results indicate that nickel adsorption on EZ and FEZ is a multimolecular layer, spontaneous, and endothermic process. Concomitantly, the desorption results reflect the high reusability of these two nanomaterials, collectively suggesting the use of waste in the design of new, low-cost, and highly efficient composite nanoadsorbents for environmental bioremediation.

## 1. Introduction

Heavy metal removal is a major concern in wastewater treatment, as their concentration is constantly rising due to human activities. In fact, the presence of heavy metals in industrial effluents, as a result of the ongoing intensive global industrialization, is a real danger to human health and the environment, given their non-biodegradable nature and incessant accumulation in the human body and soil. Furthermore, climate changes have negatively affected the water cycle, influencing the quality and availability of natural reserves. In this context, it is imperative to adopt a sustainable strategy for water management, the vital source for the existence of life [1].

In the industrial world, on the other hand, modern economic growth is directly linked to metal production, prominently nickel production, due to its exceptional physicochemical properties, including high corrosion resistance and toughness, resistance to broad temperature variations, and unique magnetic and electronic properties. In such a context, nickel plays an essential role in the energy industry, transport, pigments, medicine, tannery, food, construction, low-carbon technologies, and batteries for electric cars. However, nickel production and the life cycle of nickel-based materials are associated with a negative environmental impact [2,3,4]. In fact, nickel pollution is a major health hazard. Recent studies report that nickel in contaminated water exhibits high bioavailability, with nickel crossing various biological barriers in the human body (placenta, the blood–brain barrier, intestinal barrier), affecting the kidneys, liver, bones, and gut microbiota and concomitantly influencing neurobehavioral functions, the immune system, and causing testicular degeneration, human male infertility, fetal malformations, systemic contact dermatitis, and cancer [5,6]. According to the EU Council Directive 98/83/EC on the quality of water for human consumption, the maximum allowable concentration of nickel in water is 20 μg/L [7].

To handle the specific issue, different methods have been reported to remove nickel from wastewater: chemical, electrochemical (electrocoagulation), membrane filtration, adsorption, ion exchange, magnetic field, advanced oxidation, etc. [8]. Each method has advantages and disadvantages related to efficiency, operating costs, and environmental impact. However, adsorption stands out as the most appropriate process for nickel removal due to its simplicity, low operating cost, and high heavy metal removal performance [8,9]. In such a process, efficacy depends on the physicochemical properties of the adsorbent, i.e., surface area, porosity, surface reactivity, chemical and thermal stability, selectivity, and regeneration capacity [8,10,11]. Consequently, various adsorbent types (carbon-based, chitosan-based, mineral, magnetic metal–organic frameworks, biosorbents) have been developed over the years and used for heavy metal removal from wastewater [8,10,11].

In view of the aforementioned issues, water management in a sustainable economy demands simple, high-performance, eco-friendly, and cost-effective technologies, with a central pillar of this new economic system being the recovery and reuse of waste by designing innovative materials with high-added value. The implementation of these ideas, however, requires research creativity and the capability of specific materials to efficiently remove nickel, while concurrently supporting water quality and utilization [12]. Zeolites are among the appropriate materials in such a framework. Zeolites are hydrated aluminosilicates, which belong to the category of mineral adsorbents [8,10] that have a series of advantages: they are cheap, ecological, and easily accessible, and most of all, they have unique morpho-structural properties (high porosity and resistance to alteration, high ion exchange selectivity, large surface area, and bulk density). Based on such properties, zeolites have been studied in adsorption processes and demonstrated high performance in the removal of various pollutants from wastewater [8,10,13,14,15,16,17,18].

To this end, current research focuses on the adsorption properties of some materials obtained from agricultural waste of vegetable (vegetable or fruit peels, sawdust, nut shells, fruit seeds, tea leaves) or animal (animal bones, crustacean shells, eggshells) origin. Promising results in the literature suggest the possibility of developing green and cost-effective methods for sustainable water and waste management [19,20,21,22,23,24,25]. Among such adsorbents, eggshells are a good candidate material. In fact, worldwide, eggshells are produced in vast quantities that end up in landfills, where they become a culture medium for different microorganisms, attracting rodents and other parasites, collectively emerging as a health hazard and an important pollutant according to the Environmental Protection Agency [25,26]. Therefore, re-entry of this waste into an economic cycle to obtain new materials with high added value, including lactose-free dairy products, milk and calcium, biomaterials for orthopedics and dentistry, animal feed, heavy metal remediation, and fertilizers, represents a sustainable solution for both waste and water management [25,26].

Even if adsorption is the simplest, ecologically sound, and cheapest method for wastewater remediation, the main shortcomings are dictated by specific adsorbent characteristics, including adsorption rate, selectivity, and lifetime. Undoubtedly, clean water is critical for biodiversity, health, and life support. Consequently, efficient and inexpensive approaches are required for the development of cheap, eco-friendly adsorbents with high performance [8,10,27,28].

Although natural adsorbents are a more accessible and cheaper option, engineered materials ensure higher adsorption capability (increasing surface area and pore dimension), selectivity, and stability [10,27]. In that sense, research on developing engineered waste eggshells using functionalization with α-FeOOH particles reflects a substantial improvement in the adsorption capacity of eggshells [25,27]. In this study, the approach to developing new adsorbent materials is implemented on an innovative level, where two different composite nanoadsorbents from waste eggshells are prepared for immobilizing nickel that must be removed from aqueous solutions.

Functionalization of the first nanoadsorbent involves loading the eggshell with zeolite particles. The second nanoadsorbent emerges with the simultaneous loading of each component (zeolite and eggshell) with α-FeOOH particles, to ensure a considerable increase in sorption sites and surface area for heavy metal ions. Inevitably, each component of the prepared composite nanoadsorbent is ecological, cheap, and available, with high adsorption capability and cycling stability. To the best of our knowledge, this is the first study reporting on the concurrent use of zeolites and eggshells for the efficient removal of nickel from wastewater.

A systematic comparative study of temperature, initial concentration, adsorbent dose, contact time, and pH was also performed to evaluate the influence of experimental conditions on the adsorption capacity of the new nanoadsorbents. The adsorption behavior of both nanomaterials was investigated further by conducting adsorption isotherm, kinetic, and thermodynamic studies, and adsorption mechanism and desorption kinetics work. The physical and chemical characteristics of these new materials were studied thoroughly using several analytical methods, including the Brunauer–Emmett–Teller (BET) method, X-ray diffraction (XRD), Fourier-transform infrared (FTIR) spectroscopy, and scanning electron microscopy (SEM). The collective results depict a well-defined profile of newly composite nanoadsorbents that are highly performant, selective, recyclable, low-cost and eco-friendly, and appropriate for wastewater remediation. Overall, the herein-reported study provides a novel ecological strategy based on waste for efficient nickel recovery from aqueous solutions into new materials with high economic value and environmental performance, exemplified by water remediation and return of nickel to its life cycle, which is an essential metal in contemporary industry and economy.

## 2. Methods

### 2.1. Materials

All reagents were of analytical grade and purchased from commercial sources (Merck (New York, NY, USA), Alfa Aesar (Haverhill, MA, USA), Sigma-Aldrich (St. Louis, MO, USA)). They were used without further purification.

Eggshells (ESs) were collected from housework and washed four times with ultrapure water to remove any impurities. Then, they were dried in an oven at 60 °C for 3 h. Finally, they were crushed and sieved to obtain a powder of particles with a size in the range of 80 to 100 μm [21].

Zeolite was bought from Bentonita (Mediesu Aurit, Satu Mare, Romania). It was ground in a mortar and sieved through several ASTM sieves. The present study used only the particles that passed through a 0.42 mm mesh sieve. Zeolite was washed several times with ultrapure water to remove any soluble salts, dried in an oven for 24 h at 80 °C, cooled down to room temperature, and stored in a desiccator [14,15]. Composite materials composed of eggshells and zeolite (EZ) and iron(III) oxide-hydroxide, eggshells, and zeolite (FEZ) were prepared as stated below (vide infra).

Solutions of Ni(II) (1–30 mg/L) at different concentrations were prepared from a stock solution of NiCl_2_ (Merck, Darmstadt, Germany), dissolved in an appropriate volume of ultrapure water, with subsequent dilutions to the desired final concentration(s). For pH adjustment, 1 M HNO_3_ or NaOH solutions were used.

### 2.2. Instrumentation

The phase composition of the derived adsorbents and their components was determined using a Rigaku Ultima IV diffractometer equipped with a D/teX Ultra detector. The crystallite mean size was calculated with the whole powder pattern fitting (WPPF) method. FT-IR spectra of FEZ and its components in the solid phase were recorded using a Fourier transform infrared spectrometer (Spectrum 100 FT-IR, Perkin–Elmer, Waltham, MA, USA). The surface area of the nanoadsorbent and its components were measured using multi-point regression in the 0.08–0.3 relative pressure range and the (Barrett–Joyner–Halenda) BJH method, respectively, using a Nova 1200e high-speed surface area and porosity analyzer (Quantachrome, Boynton Beach, FL, USA). Morpho-structural analysis of the nanoadsorbents was conducted using an SEM-EDS system (QUANTA INSPECT F50, Eindhoven, The Netherlands) equipped with a field emission gun (FEG). A Jaluba SW23 thermal shaker was used for the batch adsorption experiments. A planetary Fritsch Pulverisette mill was used to prepare the new composite nanomaterials. The initial and residual concentrations of heavy metals were determined using an atomic absorption spectrophotometer (Varian SpectrAA 280 FS adsorption, Varian, Palo Alto, CA, USA).

## 3. Preparation of Adsorbents

### 3.1. EZ Nanoadsorbent

To prepare the EZ nanoadsorbent, zeolite and eggshells were mixed in a 1:1 mass ratio. Then, they were mechanically milled using a planetary mill Fritsch Pulverisette mill at 500 rpm for 15 min at 22 °C.

### 3.2. FEZ Nanoadsorbent

A 1.0 M Fe_2_(SO_4_)_3_ solution and a 2.0 M NaOH solution were added dropwise in a flask, under continuous stirring, until the pH of the mixture was 11.6. The emerging suspension was incubated at room temperature (23 °C) for 48 h to obtain iron(III) oxide-hydroxide (α-FeOOH). Subsequently, the obtained product was washed with distilled water until the pH decreased to ~6. Then, FEZ was prepared in distilled water (pH 6) from an EZ and α-FeOOH suspension in an EZ:α-FeOOH = 2:0.25 mass ratio. The resulting mixture was shaken at room temperature for 24 h. Subsequently, it was washed with distilled water several times, filtered, and dried at 70 °C for 48 h, thereby affording the composite nanoadsorbent material FEZ.

### 3.3. Batch Adsorption Study

The adsorption behavior and mechanism of action for EZ and FEZ were studied using various isotherm, thermodynamic, and kinetics models.

#### 3.3.1. Kinetic Study

Kinetic parameters were evaluated to monitor the extent of heavy metal removal. The effect of adsorbent quantity (0.50–3.5 g), contact time (0–460 min), pH (3–9), Ni(II) initial concentration (1–30 mg/L), and temperature (0–50 °C) on nickel adsorption kinetics was systematically investigated. Batch tests were conducted in 150 mL Erlenmeyer flasks containing 50 mL of the metal ion solution with a fixed initial concentration. The flasks were placed on a thermostat shaker at (21.5 °C) and a 180 rpm steady contact time, with condition variables including pH, adsorbent mass, and temperature of the experiment, until the adsorbate concentration reached equilibrium. The adsorbent from the emerging suspensions was removed by centrifugation, followed by filtration through Whatman filter paper (0.45 μm). Subsequently, the concentration of nickel in the filtrate was determined using atomic absorption spectrophotometry. Each experiment was repeated three times. The obtained data and results were accurate to 2%.

The amount of Ni(II) uptake by the adsorbent at equilibrium, Qe (mg/g), was calculated using the following equation (Equation (1)):(1)$$\mathrm{Qe}=\frac{(C_{0}-C_{e})V}{M}\;(\mathrm{mg}\;\mathrm{Ni}/g)$$

Removal efficiency (Re%) was determined as (Equation (2)):(2)% Re=(C0−Ce)×100C0
where *V* (mL) represents the volume of the solution, *M* (g) is the weight of the dry adsorbent, and *C*_0_ and *C_e_* (mg/L) are the liquid phase concentrations of nickel initially and at equilibrium, respectively.

#### 3.3.2. Adsorbent Performance

The performance of the prepared adsorbent(s) was evaluated with respect to each one of its components and monitored according to contact time. The experimental procedure was as follows: Erlenmeyer flasks (150 mL) containing 2.00 g of adsorbent, with a constant volume of nickel solution (50 mL; 25.5 mg/L) at pH 7, were kept at room temperature (22 °C) and 200 rpm. During the experiment, samples were collected at different times (0–720 min), centrifuged, and then filtered through Whatman filter paper (0.45 μm). The residual nickel concentration was determined using atomic absorption spectrophotometry [29].

#### 3.3.3. Desorption Study

Experiments were carried out by incubating samples at room temperature (22 °C), containing a constant volume (50 mL) of metal solution (25.5 mg/L) with a fixed amount (2.00 g) of the used adsorbents in 10 mL of three different solutions (0.1 M HNO_3_, 0.1 M HCl or 0.1 M NaOH). The Erlenmeyer flasks were shaken at 200 rpm. Test samples were collected every 10 min over the duration of the experiment (0–720 min), centrifuged, and then filtered through Whatman filter paper (0.45 μm). The desorbed amount of nickel was determined using atomic absorption spectrophotometry.

The desorption rate was calculated according to the following equation (Equation (3)):(3)$$\%\;D=\frac{C_{d}}{C_{a}}\times 100$$
where

*C_d_* = amount of metal ion desorbed;

*C_a_* = amount of metal ion adsorbed.

To test the FEZ adsorbent reusability, adsorption–desorption cycles were repeated 13 times on the same sample of adsorbent recovered using a 0.1 M HNO_3_ solution [16].

The experimental procedure was as follows: To a fixed amount of adsorbent (2.00 g) mixed with a constant volume (50 mL) of nickel solution (25.5 mg/L), 25 mL of HNO_3_ (0.1 M) was added. The mixture was shaken at 200 rpm at room temperature (22 °C) for 5 h, centrifuged, and then filtered (0.45 μm). The residual nickel concentration was determined using atomic absorption spectrophotometry.

#### 3.3.4. Kinetic Studies

Experiments were carried out at constant temperature (40 °C) and pH value (pH 7) with 2.00 g of absorbent and 50 mL nickel solution (25.5 mg/L). The samples were retrieved at different times (0–360 min) [30].

#### 3.3.5. Thermodynamic Study

Experiments were carried out at three different temperatures (295.15 K, 303.15 K, and 313.15 K) at pH 7 using a fixed amount of adsorbent (2.00 g) and a constant volume of nickel stock solution (50 mL; 25.5 mg/L). Adsorption thermodynamic diagrams were generated by plotting lnK (abscissa) vs. 1/T (ordinate). The correlation coefficient R^2^ = 0.9996 (FEZ) and R^2^ = 0.9994 (EZ) demonstrated a good linear relationship of the derived data.

#### 3.3.6. Statistical Analysis

Each experimental set was performed in triplicate, using one-way analysis of variance (ANOVA) without replication; *p* < 0.05 is taken as statistically significant. The BET analysis was performed using the statistical test Two-Sample *t*-Test: Assuming Equal Variances (Excel, 2013).

## 4. Results

### 4.1. BET Analysis

The surface properties of EZ, FEZ, and their component materials (zeolite and eggshell) were examined using low-temperature (77 K) nitrogen adsorption–desorption isotherms. The surface areas and pore size distributions were determined using the Brunauer–Emmett–Teller (BET) and Barrett–Joyner–Halenda (BJH) methods. The obtained results are presented in Table 1.

To check for differences between samples, the variance in surface areas, average pore size diameters, and total pore volumes were compared using the two-sample assuming equal variances (Excel, 2013) statistical *t*-test. The observed differences were statistically significant (*t* stat < *t* Critical). As a null hypothesis, the test assumes equal variances in the investigated samples, a hypothesis which is rejected if *t* Critical is higher than *t* Stat.

According to the data, the eggshell BET/N_2_ specific surface is 1.311 m^2^/g, which is similar to that reported in the literature [21,30]. The corresponding value for zeolite is 12.111 m^2^/g, analogous to that in the literature [15]. The textural properties of FEZ are different from its components, i.e., eggshell and zeolite. Thus, the specific surface of FEZ is 23.901 m^2^/g.

The isotherms of FEZ (Appendix A) fitted a type II isotherm with an H3 hysteresis loop and a type IV isotherm with an H3 hysteresis loop for the EZ adsorbent, zeolite, and eggshell (mesoporous structure) [31].

The significant increase in the surface area of EZ and FEZ nanoadsorbents, which exceeds 50%, compared with plain eggshell and zeolite particles, represents an excellent improvement in the design of new materials for nickel adsorption. This improvement, however, comes at the expense of a decrease in the average pore diameter and total pore volume compared with zeolite, resulting from the presence of eggshell particles in both EZ and FEZ. On the other hand, these contemporaneous changes provide excellent versatility in choosing the right material for a given application.

### 4.2. FT-IR Spectroscopy

FT-IR spectra were recorded for EZ, FEZ, and their starting material components (zeolite and eggshell) (Figure 1 and Appendix A). The spectrum for EZ shows eggshell vibrational bands at ~714 cm^−1^ (Ca-O stretch), 873 and 1423 cm^−1^ (C-O stretch), 1802 and 2519 cm^−1^ (attributed to O-C-O), 1643 cm^−1^ (assigned to N-H), and 2976 cm^−1^ (symmetric and antisymmetric C–H stretching vibrations) [21,27,29,32]. The weak band at ~1387 cm^−1^ is likely due to nitrate impurities from the KBr pellet [33].

The peaks associated with the zeolite component are found at 3625 cm^−1^, attributed to Si-OH-Si or Al-OH-Al, ~1057 and 797 cm^−1^, corresponding to Si-O stretching vibrations in quartz. Those at 608 cm^−1^ are assigned to Si-O-Al and Si-O-Si bending vibrations, with features at 469 cm^−1^ being attributed to Si-O-Si vibrational deformation [15,34].

The FEZ spectrum exhibits eggshell characteristic peaks at 714 cm^−1^ (associated with Ca-O stretch), 872 and 1420 cm^−1^ (C-O stretch), 1802 and 2517 cm^−1^ (attributed to O-C-O), 1647 cm^−1^ (assigned to N-H), and 2976 cm^−1^ (symmetric and antisymmetric C–H stretching vibration) [21,27,29,32]. Features associated with the zeolite component are found at 3624 cm^−1^, attributed to Si-OH-Si or Al-OH-Al; ~1055 cm^−1^, corresponding to Si-O stretches in quartz; 608 cm^−1^, attributed to a bending vibration in Si-O-Al and Si-O-Si; and 469 cm^−1^, assigned to the Si-O-Si vibrational deformation [15,34].

It appears that the EZ spectrum displays the functional groups of its components (eggshell and zeolite), thus demonstrating the successful preparation of the material. Furthermore, the stretching vibrations in Fe–OH (450, 410 cm^−1^), Fe–O (632 cm^−1^), and a feature at 797 cm^−1^, associated with Fe-O-OH bending vibrations, reflect the presence of α-FeOOH particles on eggshell and zeolite, thus pointing to a successful preparation of FEZ [35].

### 4.3. SEM Analysis

The surface morphology, shape, and particle size of both proposed nanoadsorbents (EZ and FEZ), as well as their components (eggshell and zeolite), were studied using SEM (Figure 2).

The eggshell micrographs (Figure 2A–C) appear to indicate the presence of multiporous and irregular shape agglomerations of different size particles (average size ~80 nm) [27,36]. The zeolite micrographs (Figure 2D–F) exhibit agglomerations of cubic and rectangular crystals of nanometric dimensions (~20 nm). The EZ micrographs (Figure 2G–I) indicate the presence of clusters of particles of different sizes in the nano-size regime, cubic, rectangular-shaped crystals, and irregular crystal structures loaded in the pores of eggshell particles. The FEZ micrographs (Figure 2J–L) show the same clusters of different nano-size particles (~17.4 nm) as in EZ. Nonetheless, a notable difference appears specifically in the cluster size decrease. Another visible aspect is the presence of numerous uniform nano-size particles (~7 nm) loaded in zeolite and eggshell pores that could be attributed to α-FeOOH particles [27].

Accompanying the SEM spectra are EDX analyses on the elemental composition of all samples investigated (Appendix A). The EDX spectra of eggshell samples are in good agreement with data reported in the literature [26], and the zeolite data also corroborating those in the literature [33]. The work performed on the emerging EZ nanoadsorbent material, using SEM and EDX analysis, indicates the presence of both eggshell and zeolite components (Figure 3). Analogous work performed on FEZ shows similar behavior with the introduction of the ternary component of iron (Appendix A).

In fact, the EDX spectrum for FEZ (Appendix A) shows that the iron peaks are much more intense than in EZ (Figure 3), in which the iron peaks come only from zeolite (Appendix A). Comparative analysis of live maps for FEZ (Appendix A) and EZ (Appendix A) shows differences in the identified element ratio for these two adsorbents due to the functionalization with α-FeOOH.

The collective results suggest that functionalization of EZ with FeOOH led to a new nanoadsorbent material with a unique structure, i.e., the double functionalization of eggshells with the nano-size particles of zeolite was achieved simultaneously upon loading of the pores on the eggshell and zeolite surfaces with α-FeOOH particles. The EZ nanoadsorbent structure modification using functionalization with α-FeOOH led to the active surface increase, an aspect confirmed with BET analysis (Table 1). The adsorbent surface enhancement indicates that a higher number of sorption sites are available, suggesting improvement in adsorption performance. A schematic representation of both adsorbent structures (EZ and FEZ) is presented in Figure 4.

Under the experimental conditions, the amount of eggshell/zeolite (EZ) powder exceeded that of α-FeOOH, i.e., eight times higher, along with the increased reaction time. Thus, α-FeOOH was immobilized into the EZ matrix. On the basis of the physicochemical properties of the starting reagents eggshell/zeolite (EZ) and α-FeOOH used in the preparation of FEZ composites, it is likely that ion exchange and electrostatic interactions, inherent to the nature of the components in eggshell and zeolite (EZ), further assisted by dipolar interactions with α-FeOOH in an aqueous medium (not discounting hydrogen bonding interactions), facilitate immobilization of α-FeOOH over the eggshell/zeolite surface, thereby giving rise to the assembly of the ternary composite FEZ [37,38].

### 4.4. XRD Study

The XRD spectrum of zeolite (Appendix A) shows the diffraction peaks of the crystalline phase Al_6.97_Ba_0.33_Ca_1.57_K_0.57_Mg_0.72_Na_1.92_O_96.41_Si_29.04_ (clinoptilolite-Ca) (database card no. 9001509), with a crystallite mean size of 24.7 nm. The corresponding spectrum of eggshell (Appendix A) shows diffraction peaks of the single crystalline phase of calcite CaCO_3_ (database card no. 9007689), with a crystallite mean size of 87.2 nm [21,36].

In the XRD spectrum of EZ (Figure 5 and Appendix A), only two crystalline phases, i.e., calcite CaCO_3_ from eggshell and Al_6.97_Ba_0.33_Ca_1.57_K_0.57_Mg_0.72_Na_1.92_O_96.41_Si_29.04_ (clinoptilolite-Ca) from zeolite, are visible. This result confirms the nature of the EZ absorbent. The XRD spectrum of FEZ (Figure 5 and Appendix A) shows only two crystalline phases: clinoptilolite-Ca (Al_6.97_Ba_0.33_Ca_1.57_K_0.57_Mg_0.72_Na_1.92_O_96.41_Si_29.04_ (database card no. 9001509) and calcite (CaCO_3_, database card no. 9007689). Although according to the synthesis procedure and EDX results (Appendix A), iron accounts for a large proportion in the material (mass ratio EZ:FeOOH = 2:0.25) and therefore it should be in the form of α-FeOOH, iron does not appear in the spectrum. This is likely due to the amorphous form of α-FeOOH, and as a result, it does not appear in the XRD spectrum of FEZ. The FEZ material was calcined at 300 °C and 700 °C to transform α-FeOOH from the amorphous to the crystalline Fe_2_O_3_ phase in order to demonstrate the presence of α-FeOOH and implicitly validate the FEZ preparation. After heating the material at 300 °C, the spectrum shape remained identical, showing the same crystalline phases as prior to heating.

The spectrum of the material heated at 700 °C (Figure 6) is entirely different. The diffraction peaks of clinoptilolite-Ca are considerably smaller. Only the most intense ones are still visible. The calcite peaks disappeared due to CaCO_3_ decomposition (red lines under the XRD peaks; Figure 6) into CaO and CO_2_, as proven by the presence of calcium oxide peaks (CaO, database card no. 7200686, green lines under the XRD peaks; Figure 6). The amorphous FeOOH decomposition led to the formation of crystalline hematite (Fe_2_O_3_, database card no. 9015065, brown lines under the XRD peaks; Figure 6), whose peaks are visible in the XRD spectrum [39]. The XRD results demonstrate that the FEZ nanoadsorbent was successfully prepared.

### 4.5. Adsorption Properties

#### 4.5.1. Effect of Adsorbent

The nickel removal efficiency and adsorption capacity of EZ and FEZ were examined as a function of adsorbent mass (Figure 7A,B).

The graphs in Figure 7 show an increase in nickel adsorption when the adsorbent amount rises from 0.50 g to 2.0 g. Adsorption reaches its maximum at 2.0 g of adsorbent (99.9% and 321.1 mg/g for FEZ, and 97.3% and 287.9 mg/g for EZ, respectively). After reaching equilibrium, adsorption shows a slight downward trend with increasing amounts of adsorbent. The results indicate that increasing adsorbent mass ensures greater availability of active sites until equilibrium is reached. After that, boosting adsorbent mass leads to agglomeration, thus decreasing the specific surface area and the active sites [15,40,41,42].

#### 4.5.2. Effect of Initial Concentration on Nickel Removal Efficiency

The initial pollutant concentration represents one of the main driving forces in the adsorption process. To that end, the effect of the initial heavy metal concentration on the nickel removal efficiency and adsorption capacity was investigated (Figure 8A,B).

It can be seen that the removal efficiency shows direct proportional dependence on the increase in the initial concentration of the pollutant, in the range of 0–30 mg/L, for both nanoadsorbents (FEZ, EZ) (Figure 8B). Figure 8A shows that the adsorption capacities for FEZ and EZ exhibit an increasing trend in the same range of the initial pollutant concentration, 0–26 mg/L, reaching a maximum of 321.1 mg/g for FEZ and 287.9 mg/g for EZ.

Maximum removal efficiencies for FEZ (99.9%) and EZ (97.3%) were obtained at Ni(II) concentrations of 25.5 mg/L (Figure 8B). Past that point, both removal efficiencies follow a slightly decreasing trend. The same trend is observed for the adsorption capacities. According to collision theory, these results indicate that an increase in nickel concentration (and implicitly, the number of nickel ions) leads to an increase in the reaction rate due to numerous possibilities of interaction with acceptor sites on FEZ and EZ until the equilibrium concentration has been reached [43]. Past the equilibrium point, an imbalance between a large number of nickel ions and a progressively decreasing number of active sites available on the adsorbents causes a decrease in the adsorption potential for both FEZ and EZ. The obtained results corroborate the data reported for the component materials of the adsorbents [21,40,44].

#### 4.5.3. Effect of pH

pH has a dominant effect on the adsorption process because it influences the degree of ionic chemical speciation of the adsorbing species and adsorbent surface [15,27,41,45,46,47]. In view of the aforementioned information, nickel ion adsorption on FEZ and EZ was investigated as a function of pH. To that end, the relationship between pH and nickel removal efficiency, as well as adsorption capacity, are shown in Figure 9A,B.

The results show that an increase in pH from 3.0 to 7.0 induces a significant increase in the adsorbed nickel ions per unit mass of adsorbent. The adsorption efficiency and adsorption capacity reach a maximum value (99.9% and 321.1 mg/g for FEZ, and 97.3% and 287.9 mg/g for EZ) at pH 7.0, after which no further changes occur. It appears, therefore, that in an acidic environment, there is competition between protons and nickel cations for active sites available in the adsorbent. Moreover, boosting positive charge density on the adsorbent surface induces an electrostatic repulsion force on nickel ions. Consequently, the adsorption rate is low in an acidic environment. When the pH increases, both the competing effect of protons and the electrostatic repulsion forces decrease, with the nickel ions increasingly occupying active sites in the adsorbent, thus increasing nickel removal efficiency [15,27,41,45,46,47].

At pH > 7.0, the generation of hydroxide ions prevails, with [Ni(OH)]^+^ being the dominant species causing deceleration in the metal ion removal rate. Therefore, pH 7 was selected as the optimal value for further experiments. The emerging results are in good agreement with the reported data for the starting component materials used in adsorbent preparation [15,27,37,41,45,46,47].

#### 4.5.4. Effect of Contact Time

The time required for the adsorption of a pollutant is a critical factor, on which the cost of the adsorption process depends heavily [15,40,41,48]. Nickel ion uptake capacities were determined as a function of contact time to establish an optimum contact time, at which the adsorption capacity and pollutant removal efficiency are maximized for each of the two nanoadsorbents studied. The results (Figure 10A,B) show that removal efficiency and adsorption capacity increase with increasing contact time for both the investigated adsorbents.

Moreover, equilibrium is reached at 240 min, with a maximum adsorption capacity of 321.1 mg/g for FEZ and 287.9 mg/g for EZ. Also, at this time point, nickel removal efficiency reaches a maximum of 99.9% for FEZ and 97.3% for EZ. Further careful observation of the data suggests that there are three clear stages, in which nickel adsorption takes place:(a)In the first stage (0–120 min), adsorption increases rapidly, a phenomenon that could be attributed to the high availability of active sites in the adsorbent;(b)In the second stage (120–240 min), attenuation of the adsorption rate occurs, as a result of the decrease in available active sites;(c)In the third stage (240–460 min), metal adsorption exhibits a plateau trend, indicating saturation of the active sites, after reaching equilibrium. The data suggest that the optimal time necessary for the adsorption process to reach equilibrium for each of the adsorbents is 240 min.

#### 4.5.5. Effect of Temperature on the Adsorption Process

Temperature is a significant parameter in the adsorption process that influences the performance of an adsorbent [46]. Nickel uptake by FEZ and EZ adsorbents was investigated in the temperature range of 5–50 °C (Figure 11A,B).

The results show that adsorption is endothermic. Thus, nickel removal efficiency and adsorption capacity increase almost linearly with increasing temperature up to a maximum, after which there is a slight decrease with increasing temperature. The maximum removal efficiency is reached at 40 °C (99.9% for FEZ and 97.3% for EZ). At this temperature, a maximum nickel adsorption capacity of 321.1 mg/g for FEZ and 287.9 mg/g for EZ are observed. The data indicate that this temperature range is advantageous for increasing the mobility of metal ions and, implicitly, the interaction with acceptor sites on the adsorbent. To that end, the effect of temperature on the nickel adsorption process suggests that in the 5–40 °C range, adsorption is an endothermic process in which physical adsorption takes place. At temperatures higher than 40 °C, the chemisorption process occurs. However, it should be noted that even at 50 °C, very high removal efficiency values (>90%) are obtained for both nanoadsorbents (97.9% for FEZ and 92.2 for EZ), thereby suggesting that a temperature increase beyond 40 °C has a minimal effect on the adsorption process.

#### 4.5.6. Nickel Removal Efficiency—Comparative Analysis between the Adsorbent and Starting Materials

A comparison between FEZ Ni(II) removal efficiency with EZ (before functionalization with α-FeOOH) and the starting materials (eggshell and zeolite), as a function of the contact time, was carried out. Figure 12 shows that nickel removal efficiency increases with contact time for all four adsorbents: FEZ, EZ, eggshell, and zeolite.

The maximum efficiency occurs at four hours, thus pointing to adsorption equilibrium. The experimental results show that pollutant removal efficiency decreases in the following order: FEZ (99.9%) > EZ (97.3%) > eggshell (94.3%) > zeolite (88.3%). After equilibrium is reached, the adsorption rate decreases with increasing contact time. The data confirm the fact that adsorbent performance depends on the specific surface and porosity (BET analysis in Table 1). The adsorption efficiency values for the eggshell and zeolite samples are similar to those reported in the literature [15,34,45,49].

#### 4.5.7. Adsorption Isotherms

Adsorption isotherms were used to analyze the nickel partition between the adsorbent and solution at equilibrium. The Langmuir and Freundlich models are the most common and reliable models to determine the maximum nickel adsorption capacity using adsorption isotherms [21,50,51]. Several studies have reported that nickel adsorption on eggshell, clinoptilolite, and zeolite adsorbents fit Langmuir and Freundlich adsorption isotherms [13,14,18,21,52]. In addition, various studies have reported that the adsorption isotherms of Ni(II) on eggshell and zeolite (the primary materials from which the FEZ and EZ adsorbents were prepared) obtained using the mathematical equations of Temkin and Dubinin–Radushkevich (D-R) adsorption models, do not fit well the experimental results [11,13,21,26,40,49,52]. Therefore, Langmuir and Freundlich adsorption isotherms were considered to create a suitable adsorption model adequate to reproduce the experimental results of this study [50].

The Langmuir model is based on the theoretical principles that (i) the adsorbent has a single, homogeneous layer in which the adsorption process takes place, and (ii) each of the adsorbed molecules has the same adsorption energy without contact between these molecules [40,50].

The linear form of the Langmuir model is expressed using the following equation (Equation (4)):(4)$$\frac{C_{e}}{Q_{e}}=\frac{1}{kQ_{m}}+\frac{C_{e}}{Q_{m}}$$

In the Langmuir equation, the characteristic adsorption behavior can be calculated according to the following equation (Equation (5)):(5)$$R_{L}=\frac{1}{1+kC_{0}}$$
where *k* (L/mg) is the Langmuir adsorption constant.

The Freundlich model is suitable to describe the multilayer adsorption process on the heterogeneous surface with non-uniform dispersion of heat and the interaction between the adsorbed molecules [21,27]. In fact, the Freundlich isotherm was used to describe Ni(II) adsorption on the FEZ and EZ adsorbents, taking into consideration the distribution of energy sites and the competition between different ions for the adsorbent active sites available [51,53,54].

The linear form of the Freundlich model is given in the equation below (Equation (6)):(6)$$\log\;Q_{e}\;=\log\;k_{f}+\frac{1}{n}\;\log\;C_{e}$$
where *k_f_* (mg/g) and *n* (g/L) are the constants of the Freundlich isotherm.

The plotted experimental data, using the linearized form of Langmuir and Freundlich models, are shown in Appendix A.

The calculated parameters of the two isotherms are presented in Table 2.

Analysis of the isotherm data in Table 2 indicates that both models provide satisfactory descriptions of nickel absorption on FEZ and EZ. The calculated maximum values of the adsorption capacity (Q_m_) for the two considered nanoadsorbents (FEZ and EZ) are in excellent agreement with those established experimentally at equilibrium (Q_e,exp_) for the components of the newly prepared nanomaterials [21,49,53,54,55]. The applicability of these two equilibrium models in describing the adsorption process for each of the two prepared adsorbents, EZ and FEZ, was evaluated based on the value of the correlation coefficient, R^2^. The R^2^ value obtained using the Freundlich model is slightly higher than that derived using the Langmuir model, suggesting that nickel ion adsorption is a multimolecular layer adsorption process on irregular surfaces [15,18].

The values of the equilibrium parameter from the Langmuir isotherm, R_L_, were found to be in the range of 0 < R_L_ < 1 for EZ and FEZ, thus indicating a favorable adsorption process [13,50,56]. In addition, according to the data in Table 2, the values of the Freundlich constant n, which provides information about the linearity of the adsorption, are higher than 1, thus suggesting a suitable physical adsorption process occurring on the investigated EZ and FEZ nanoadsorbent heterogeneous surfaces [26,50,56].

#### 4.5.8. Thermodynamic Study

The thermodynamic profile ascertaining the feasibility of EZ and FEZ as adsorbents for nickel removal was investigated. Assessment of the adsorbent thermodynamic behavior included the following parameters: Gibbs free energy (ΔG^o^), entropy (ΔS^o^), and enthalpy (ΔH^o^), determined according to the Gibbs–Helmholtz and Van’t Hoff equations (Equations (7) and (8)):(7)$$∆G^{o}=-\mathrm{RT}\;\mathrm{lnK}$$
(8)$$lnK=\frac{-∆H^{o}}{RT}+\frac{∆S^{o}}{R}$$
where

K (mL/g) = adsorption equilibrium constant;

ΔG^o^ (kJ/mol) = free energy variation of the adsorption process;

ΔH^o^ (kJ/mol) = the standard enthalpy variation;

ΔS^o^ [J/(mol K)] = standard entropy variation;

R = 8.314 J/(mol K) = the gas constant;

T (K) = the absolute temperature.

Experiments were run at three different temperatures (295.15 K, 303.15 K, and 313.15 K), at constant pH 7, and using a 25.5 mg/mL nickel stock solution.

A Van’t Hoff’s plot for the adsorption of nickel on FEZ and EZ is shown in Appendix A. The slope and intercept correspond to the thermodynamic parameters ΔH^o^ and ΔS^o^ (Table 3).

Negative ΔG^o^ values indicate thermodynamic feasibility and spontaneity of the prepared adsorbents for nickel removal in the used temperature range. ΔH^o^ provides information on the physical or chemical nature of the adsorption process (28.89 kJ/mol for FEZ and 24.61 kJ/mol for EZ), reflecting an endothermic adsorption process with a favorable affinity for nickel i the two nanoadsorbents [11,40,51]. Positive ΔS^o^ values denote adsorbent (FEZ and EZ) affinity for nickel ions and indicate that the adsorption could involve structural changes [11,40,51,57].

#### 4.5.9. Adsorption Kinetic Study

Several mechanisms can control an adsorption process: mass transfer, particle diffusion, diffusion control or chemical reactions. To that end, kinetic studies were performed to obtain information on adsorbent effectiveness, insight into the adsorbent process (mass transfer), and dynamic parameters of adsorption (rate, temperature) [51,58]. In that respect, a pseudo-first-order kinetic model, a pseudo-second-order kinetic model, and an intraparticle diffusion model were applied to test the experimental data on nickel adsorption for the newly prepared FEZ and EZ.

The pseudo-first-order kinetic model (Lagergren equation) assumes that the adsorption rate depends on the number of active sites available [59].

The linear form of the Lagergren equation is expressed as follows (Equation (9)):(9)$$ln\;(Q_{e}-Q_{t})=lnQ_{e}-k_{1}t$$
where *Q_e_* (mg/g) and *Q_t_* (mg/g) represent the adsorption capacities at equilibrium and at time *t*, respectively, and *k*_1_ (min^−1^) represents the rate constant of adsorption kinetics.

The pseudo-second-order model assumes that the adsorption rate depends on the existence of chemical interactions between the nickel ions and the functional groups on the adsorbent [59].

The linearized form of second-order kinetics is presented in the following equation (Equation (10)):(10)$$\frac{1}{Q_{t}}=\frac{1}{k_{2}Q_{e}^{2}}+\frac{1}{Q_{e}}\;t$$
where *k*_2_ [mg/(g min)] is the rate constant of pseudo-second-order kinetics.

The Weber and Morris intraparticle diffusion model hypothesizes that the diffusion of nickel ions through adsorbent pores influences the adsorption rate. The Weber and Morris model is described as follows (Equation (11)):(11)$$Q_{t}=k_{i}t^{1/2}+C$$
where *k_i_* [mg/(g × min^−1/2^)] is the intraparticle diffusion rate constant and C (mg/g) is a constant related to the thickness of the boundary layer.

Appendix A shows the plots of the kinetic models for nickel adsorption on FEZ and EZ.

The kinetic constants were determined from the slopes and intercepts of the plots shown in Appendix A (Table 4).

The data reveal insignificant differences between the values of the correlation coefficients for the pseudo-first-order and pseudo-second-order kinetic models. This could indicate that retention of nickel on the adsorbents is achieved through a physical and chemical adsorption process. However, it can be seen that the calculated values of the adsorption capacity at equilibrium are much closer to the experimental values obtained using the pseudo-second-order kinetics model. In other words, nickel ion adsorption on the adsorbents relies mainly on chemisorption, involving chemical bond formation between nickel ions and active sites [15,21,26,40,49]. The correlation coefficient for the intraparticle diffusion model is higher than 0.97 for both adsorbents, suggesting that intraparticle diffusion is involved in the adsorption process. However, intraparticle diffusion cannot be the only rate-limiting step because the diagram is not linear and deviates from the origin [49]. The kinetics results corroborated the data reported in the literature for the starting materials of the proposed nanoadsorbents [15,21,26,40,49].

#### 4.5.10. Insight into Adsorption

Evaluation of structural and morphological changes using FT-IR and SEM-EDX techniques was used to investigate the potential nickel adsorption mechanisms of both proposed nanomaterials. A comparative assessment between the FT-IR spectra of the proposed nanoadsorbents EZ and FEZ was performed before and after nickel removal in order to reveal changes occurring in the adsorbents.

The FT-IR spectrum of EZ after nickel retention (Appendix A) shows a shift in some adsorption peaks and the emergence of new bands. Thus, shifts to higher energies of the adsorption bands from 469 (Si-O-Si vibrational deformation) to 487 cm^−1^, 797 cm^−1^ (Si-O stretching vibration) to 802 cm^−1^, 3480 cm^−1^ (OH vibrations) to 3491 cm^−1^, and 3625 cm^−1^ (Si-OH-Si or Al-OH-Al) to 3632 cm^−1^ were observed (Appendix A). These changes can be attributed to chemical interactions between nickel ions and the corresponding functional groups. Following nickel removal, the EZ spectrum shows the appearance of adsorption bands at 487, 668, and 1449 cm^−1^. The peaks at 487 cm^−1^ and 668 cm^−1^ could be attributed to Ni–O stretching vibration modes [60,61,62,63].

Assessment of the FT-IR spectra of FEZ (Appendix A) before and after Ni(II) ion removal depicted a series of notable differences (intensity of vibrational peaks and displacement or appearance of new absorption bands). Substantial changes in the O-H features at ~3437 cm^−1^ and 1636 cm^−1^ indicate that this functional group participates in the heavy metal adsorption process [32,47,64]. An increase in the intensity of the bands at 1423 and 876 cm^−1^ (C-O stretching vibration), 2511 cm^−1^ (O-C-O), and 2976 cm^−1^ (C-H symmetric and antisymmetric stretching vibrations) can also be observed. These changes can be attributed to the interaction between nickel ions and the functional groups of the FEZ adsorbent [15].

The FT-IR spectra of FEZ after adsorption show that the peaks at ~630 cm^−1^ (Fe–O stretch), 797 cm^−1^ (attributed to Fe-O-OH bending vibration), 872 cm^−1^ (corresponding to C-O stretching vibration), 1055 cm^−1^ (assigned to Si-O stretching vibration), 2517 cm^−1^ (attributed to O-C-O), and 2974 cm^−1^ (associated with C-H symmetric and antisymmetric stretching vibration) shifted.

The new absorption bands at 472 cm^−1^ and 665 cm^−1^ can be attributed to Ni-O stretching vibration modes [58,59,62]. Collectively, the FT-IR results suggest that nickel adsorption on FEZ involves chemical bond formation [65,66].

Morphological changes (particle and pore size, shape, particle distribution) in the structure of both EZ and FEZ after nickel adsorption were also investigated using SEM-EDX (Appendix A) to corroborate the FT-IR spectroscopy results. The SEM micrograph of EZ following nickel adsorption (Appendix A–D) indicates the presence of numerous particles with irregular shapes attributed to the pollutant. Also, after nickel adsorption, the SEM image of FEZ (Appendix A–E) shows the appearance of several particles with nanometric sizes and irregular shapes. On the other hand, the SEM (Appendix A) results indicate a decrease in porosity after nickel adsorption for both proposed materials. These changes in adsorbent morphology suggest that, under the employed experimental conditions, the dissolution–precipitation phenomenon plays a dominant role in the nickel adsorption mechanism [26,49].

Since zeolites can trap positively charged species, Ni ions (such as [Ni(H_2_O)_6_]^2+^ = Ni(II) and [Ni(H_2_O)_5_(OH)]^+^ = [Ni(OH)]^+^ complex ionic species, emerging in the pH-specific aqueous solution speciation of the starting nickel reagent) could seek facile immobilization into the ternary FEZ matrix at pH 7 [37]. To that end, hydrogen bonding and electrostatic interactions among the initially physisorbed nickel ion complex species approaching the EZ binary component of the FEZ matrix in aqueous media could forge the establishment of chemical bonding with surface oxygen terminal anchors, not unlike that suggested by FT-IR [37,38]. Furthermore, the nickel ions could pass through the pores of the zeolite and be confined deep inside by steric effects (e.g., Van der Waals strain) [67]. In addition, nickel ions could also be immobilized through hydrogen-bonding-mediated and electrostatic interactions by α-FeOOH (reactivity through the oxophilic terminal anchors is abundantly present in the binary EZ and ternary FEZ surface matrix) [37,38,67] in an oxygen-rich medium, such as water, essentially reflecting chemisorption activity commensurate with the previously recognized experimental events attested to by spectroscopic, kinetic, and thermodynamic studies (vide supra).

Elemental analysis and the distribution in FEZ and EZ after nickel adsorption were examined using SEM live maps (Appendix A) and an EDX analysis (Appendix A). It appears that there are differences in the amount of nickel retained on the EZ surface compared with FEZ. In fact, the nickel identified in FEZ was much higher than in EZ. Similarly, after nickel adsorption, the elemental composition of FEZ and EZ, obtained using an EDX analysis (Appendix A), indicates the presence of a peak corresponding to the heavy metal (Appendix A), thus confirming nickel adsorption [65,68].

#### 4.5.11. Comparison of Nickel Removal Efficiency among Other Adsorbents

An adsorption performance comparison between the two prepared nanoadsorbents and others reported in the literature for nickel removal is presented in Table 5.

The results indicate that the two new nanoadsorbents EZ and FEZ have a much higher adsorption efficiency than any other known materials. This could be attributed to the higher surface area compared to their components (eggshell or zeolite), following adsorbent surface modification.

#### 4.5.12. Desorption Study and Adsorbent Regeneration

Cycling stability is essential to high-performance and economic feasibility for an adsorbent [8]. Regeneration efficiency depends on the ease with which the desorption process of the adsorbed pollutant takes place [46]. Consequently, nickel ion desorption under acidic (nitric acid and hydrochloric acid) and alkaline (sodium hydroxide) conditions was investigated (Figure 13A). The results suggest that in an acidic environment, the desorption rate increases proportionally in the time range of 0–11 h, reaching the maximum value. After this point, a decrease in desorption yield with time is observed.

The desorption rate was 91% in FEZ and 89% in EZ (Figure 13A) when 0.1 M HNO_3_ was used as desorption agent. When 0.1 M HCl was used, over 80% of the heavy metal was desorbed from both nanoadsorbents. This result can be attributed to the generation of abundance of protons, thereby determining an exchange between the adsorbed nickel ions and protons and the protonation of the functional groups in the adsorbent [74]. However, in an alkaline medium, desorption rates of ~2% can be justified by the existence of two competing phenomena: nickel precipitation and the negative charge of the functional groups in the adsorbents, which favors adsorption (pH 6–9) more than nickel ion desorption [16]. To that end, thirteen cycles of nickel adsorption and desorption processes on both adsorbents were run to investigate their reusability potential. Nickel adsorption capacity variation in the two nanoadsorbents, depending on the number of adsorption–desorption cycles, is shown in Figure 13B.

After ten cycles, the adsorption capacity slightly decreased by about 12% for EZ and 10.5% for FEZ. These results indicate that the performance and reusability of the two nanoadsorbents are very high.

## 5. Discussion

We demonstrated the successful preparation of two new composite nanoadsorbents engineered from eggshell waste and zeolite (EZ and FEZ). The FEZ preparation led to double functionalization of the eggshell with the zeolite nanoparticles, achieved simultaneously by loading the eggshell pores and zeolite surface with α-FeOOH particles. The XRD spectrum of FEZ demonstrated that α-FeOOH is in an amorphous state, which could explain FEZ’s higher specific area and enhanced adsorption capacity compared with EZ. Both nanoadsorbents can be used successfully to remove nickel from aqueous solutions. Their adsorption behavior follows the Freundlich isotherm and pseudo-second-order models. The adsorption efficiency of EZ allows the removal of over 97% Ni(II) and more than 99% Ni(II) for FEZ. In contrast to the data reported for zeolite, eggshell or even functionalized eggshell, the two new materials possess superior absorption capacity attributed to the higher surface and the microporous structure resulting from the used experimental conditions. In comparison with the literature on materials similar to their components, the herein-reported new nanoadsorbents (EZ and FEZ) exhibit higher adsorption efficiency [15,21,27,49,63,65,66,68,69,70].

In addition, the materials obtained in this work show environmental performance and ensure nickel recycling in different economically valuable forms. To that end, (a) consecutively separated solutions containing nickel from the desorption process could be collected and used in nickel-plating baths, (b) the saturated EZ and FEZ adsorbents could be used as raw materials in cement or ceramic material production [75,76], and (c) the nickel-containing materials (EZ and FEZ after adsorption) could be used as fertilizers, considering the components of the newly prepared adsorbents together with the recognized role of nickel in nitrogen metabolism in plants [77,78]. Congruent with such applications is the fact that both materials show high cycling stability. After ten successive adsorption–desorption cycles, the adsorption efficiency of Ni(II) decreases by only approximately 10% for both adsorbents examined. Overall, the results reveal that these new nanoadsorbents, prepared from waste, have merit in the remediation of wastewater in the context of a sustainable economy.

## 6. Conclusions

This study describes nickel removal from aqueous solutions using two new composite nanoadsorbents, EZ and FEZ, prepared from eggshell waste, zeolite, and iron. Functionalization for EZ involved loading the eggshell pores with zeolite particles, further confirmed using SEM, XRD, and FTIR. In FEZ, however, functionalization involved simultaneous loading of each surface component (zeolite and eggshell) with α-FeOOH particles to ensure considerable increase in sorption sites and surface area for heavy metal ions. XRD demonstrated that α-FeOOH is in an amorphous state in FEZ, thus explaining the increase in the specific surface area. SEM and BET support these structural changes during FEZ preparation. Accordingly, the specific surface area increase allowed better adsorption performance for FEZ (99.9%, 321.1 mg/g) compared with EZ (97.3%, 287.9 mg/g). The best results for both adsorbents were obtained at 40 °C, a pH 7, and in 240 min. Isotherm, thermodynamic, and kinetic models indicate that nickel adsorption on the two adsorbents is achieved through physisorption events interwoven into a chemical adsorption process described with a second-order model. In addition, SEM and FT-IR studies after nickel adsorption suggest that nickel adsorption is achieved through chemical bond formation. Regeneration efficiency experiments showed that nickel could be optimally desorbed from the surface of the adsorbents using nitric acid. The maximum desorption rate was 91% in FEZ and 89% in EZ. Collectively, this study shows that highly efficient and reusable nanoadsorbents can be prepared with a simple functionalization method of eggshell waste using cheap and eco-friendly materials.

## Figures and Tables

**Figure 1 nanomaterials-13-02572-f001:**
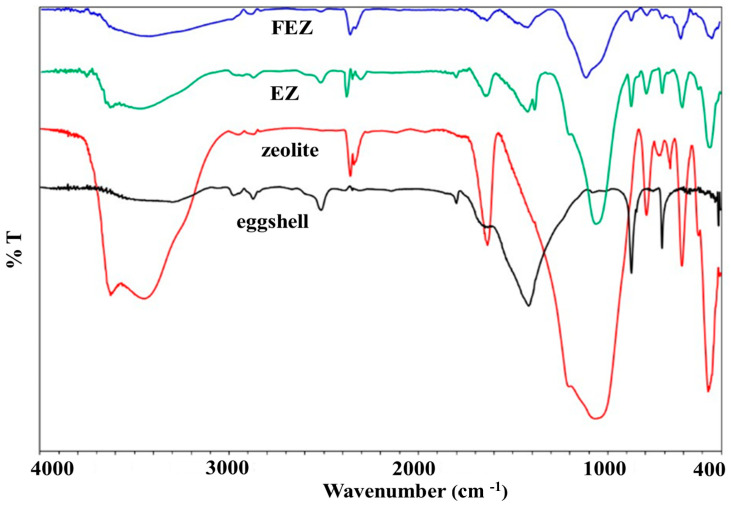
FT-IR spectra of FEZ, EZ, eggshell, and zeolite.

**Figure 2 nanomaterials-13-02572-f002:**
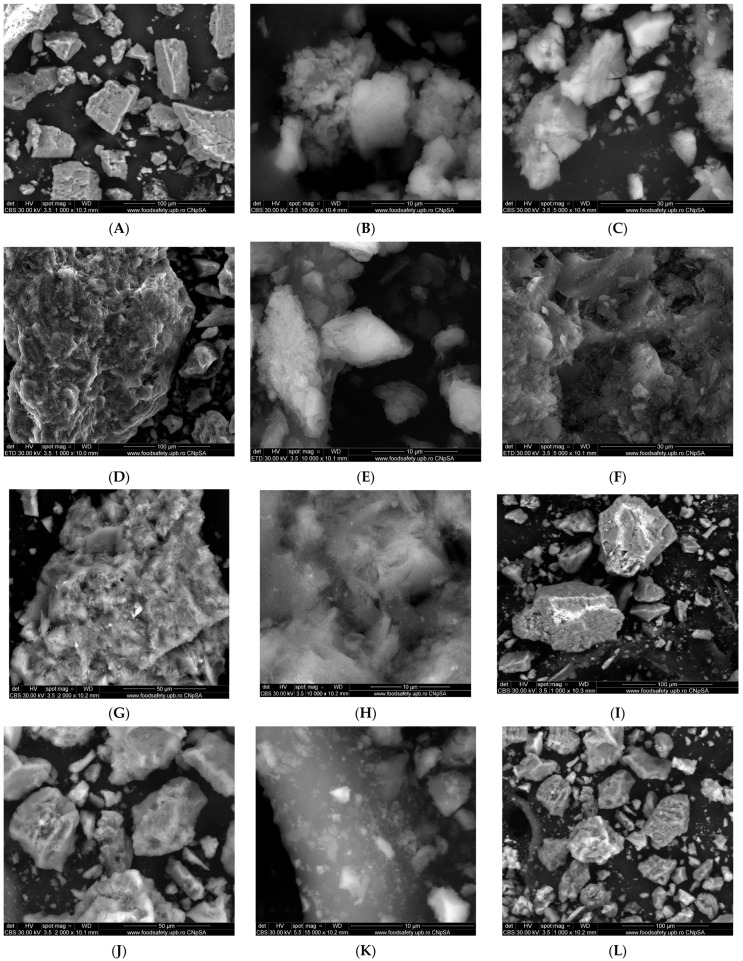
SEM images of eggshell (**A**–**C**), zeolite (**D**–**F**), EZ (**G**–**I**), and FEZ (**J**–**L**) nanoadsorbents.

**Figure 3 nanomaterials-13-02572-f003:**
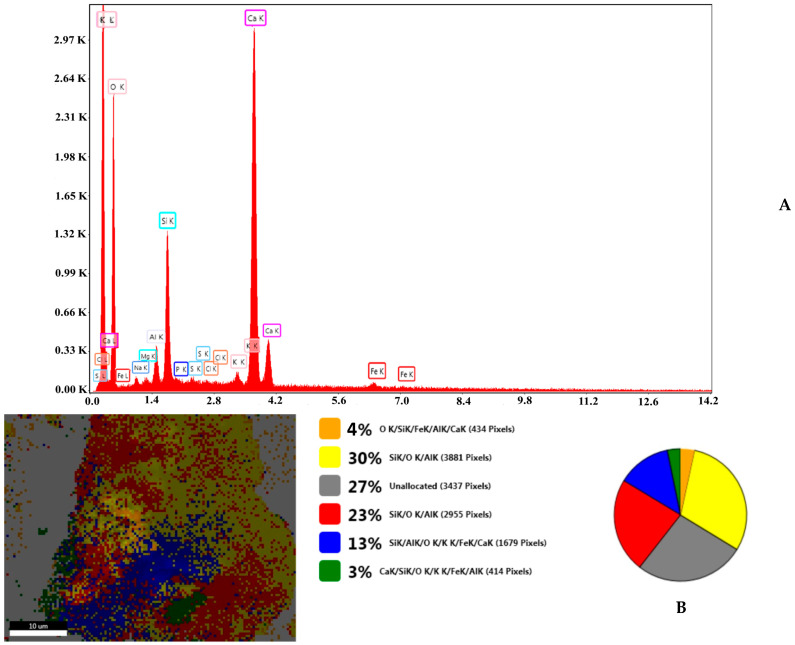
(**A**) EDX composition and (**B**) SEM-EZ live map for EZ and the distribution of the identified elements.

**Figure 4 nanomaterials-13-02572-f004:**
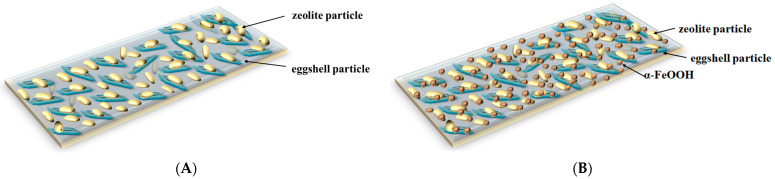
Schematic representation of nanoadsorbent structures: (**A**) EZ and (**B**) FEZ.

**Figure 5 nanomaterials-13-02572-f005:**
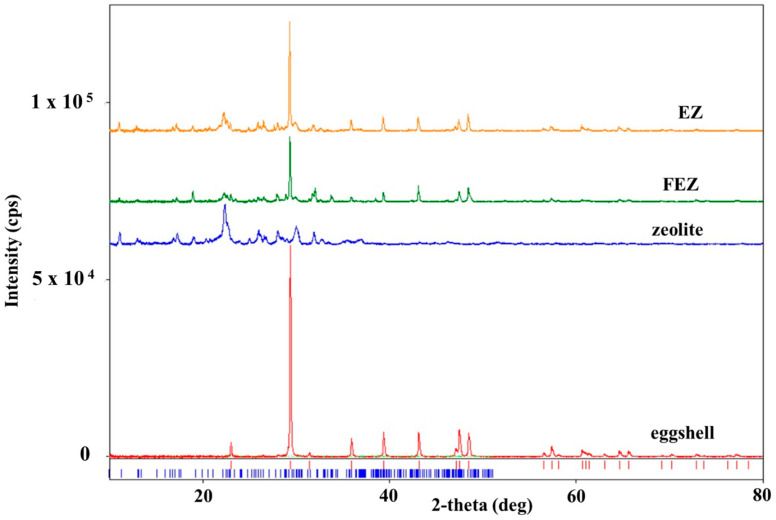
XRD spectra of eggshell, zeolite, EZ, and FEZ.

**Figure 6 nanomaterials-13-02572-f006:**
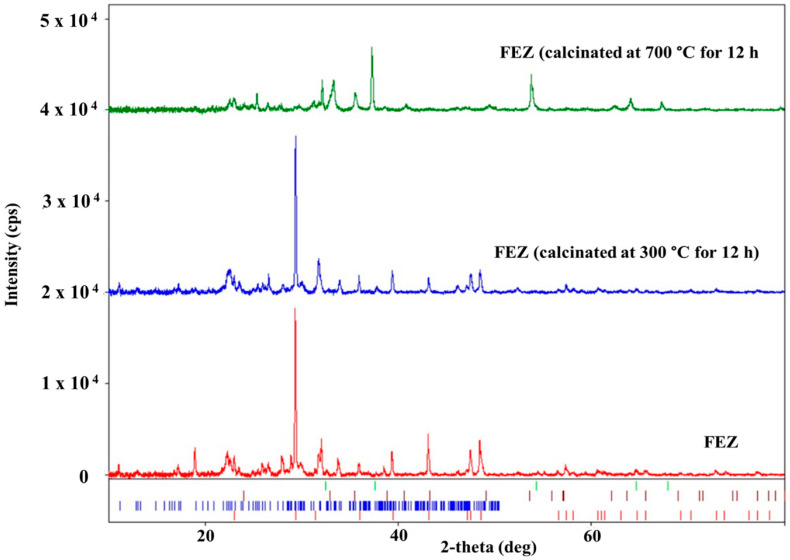
XRD spectra of FEZ before and after calcination at 300 °C and 700 °C.

**Figure 7 nanomaterials-13-02572-f007:**
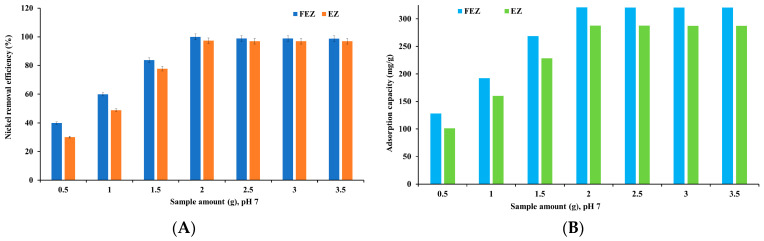
Nickel removal efficiency (**A**) and adsorption capacity (**B**) as a function of adsorbent mass.

**Figure 8 nanomaterials-13-02572-f008:**
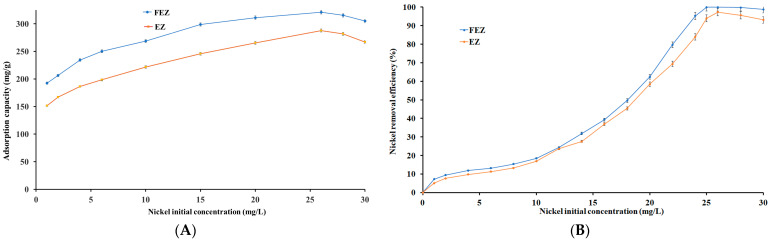
Relationship between nickel (**A**) initial concentration and adsorption capacity (mg/g) and (**B**) initial concentration and nickel removal efficiency (%).

**Figure 9 nanomaterials-13-02572-f009:**
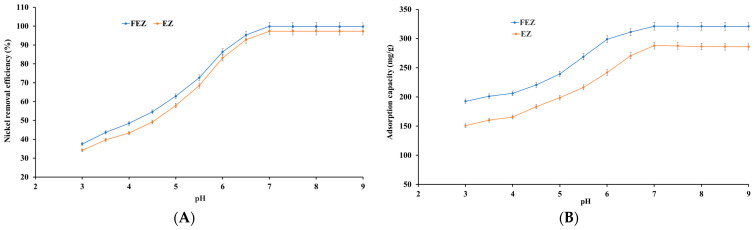
Effect of pH variation on (**A**) nickel removal efficiency and (**B**) adsorption capacity.

**Figure 10 nanomaterials-13-02572-f010:**
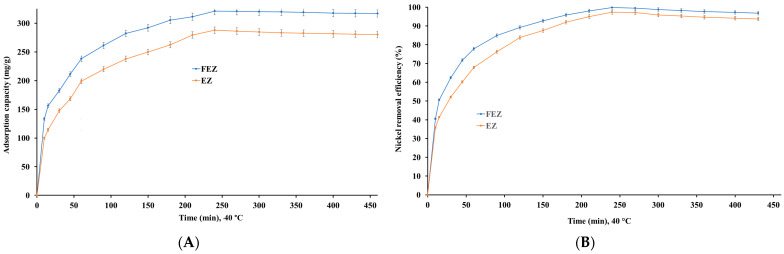
Effect of contact time on (**A**) nickel adsorption capacity, and (**B**) nickel removal efficiency.

**Figure 11 nanomaterials-13-02572-f011:**
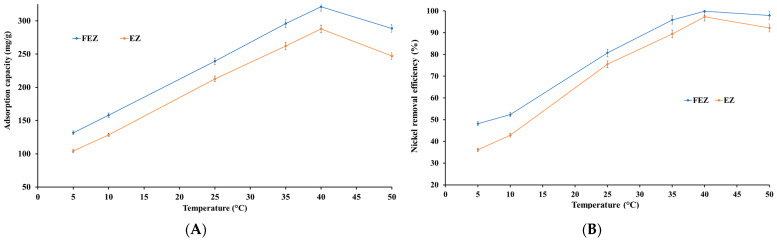
Relationship between (**A**) temperature and adsorption capacity and (**B**) temperature and nickel removal efficiency.

**Figure 12 nanomaterials-13-02572-f012:**
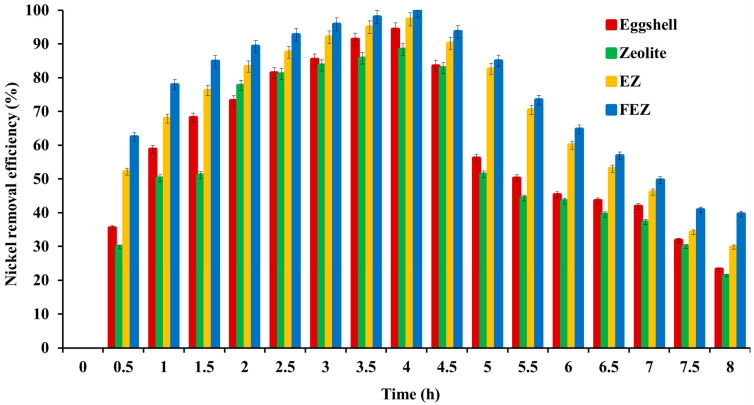
Removal efficiency and contact time relationship for all four adsorbents.

**Figure 13 nanomaterials-13-02572-f013:**
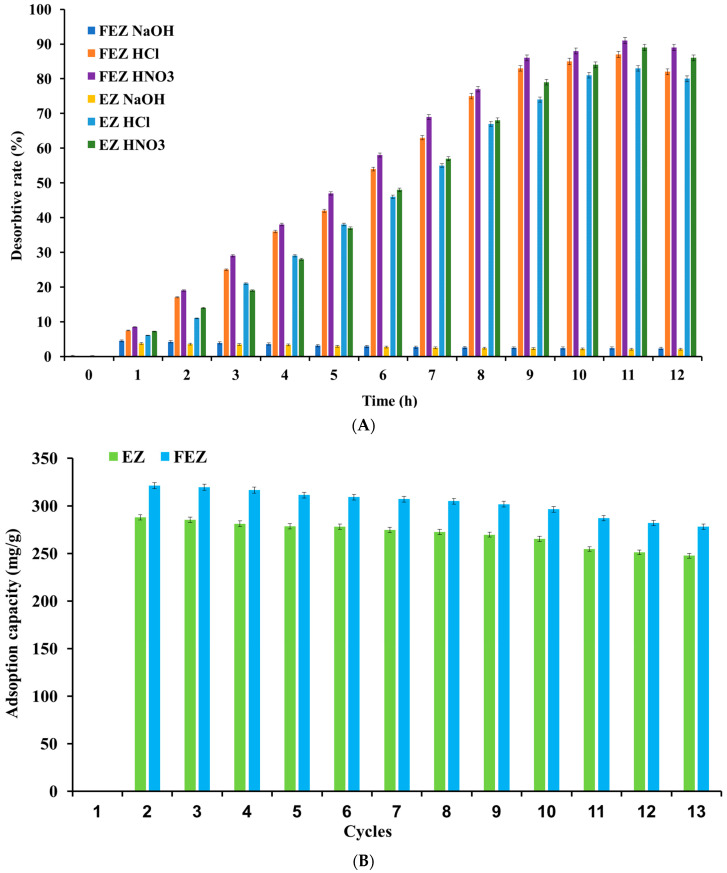
(**A**) Relationship between desorption rate and time. (**B**) Reusability of FEZ and EZ.

**Table 1 nanomaterials-13-02572-t001:** Parameters of the newly prepared nanoadsorbents and their components determined through BET *.

Sample	Surface Area (m^2^/g)	Average Pore Size Diameter (nm)	Total Pore Volume (cm^3^/g)
Eggshell	1.311	8.347	2.03 × 10^−3^
Zeolite	12.111	15.574	38.06 × 10^−3^
EZ	19.321	7.132	11.20 × 10^−3^
FEZ	23.901	4.023	8.10 × 10^−3^
*t*-test **	*t* Stat: 0.98; *t* Critical two-tail: 2.44		
		*t* Stat: −0.72; *t* Critical two-tail: 2.44	

* standard deviation (SD) = 2%. ** *t*-test: two-sample assuming equal variances (Excel, 2013).

**Table 2 nanomaterials-13-02572-t002:** The Langmuir and Freundlich model parameters for nickel adsorption on FEZ and EZ.

Adsorbent Material	Langmuir Model					Freundlich Model			
	Q_e,exp_	Q_m_	k	R_L_	R^2^	K_F_	*n*	R^2^	E_a_ (kJ/mol)
FEZ	321.1	321.0	0.283	0.861	0.9994	4.60517	1.893	0.9999	32.4
EZ	287.9	286.8	0.223	0.783	0.9989	3.85721	1.659	0.9998	32.2

**Table 3 nanomaterials-13-02572-t003:** Thermodynamic parameters for nickel adsorption on EZ and FEZ nanoadsorbents *.

T (K)	Adsorbents					
	FEZ			EZ		
	ΔG^o^ (kJ/mol)	ΔH^o^ (kJ/mol)	ΔS^o^ J/(mol K)	ΔG^o^ (kJ/mol)	ΔH^o^ (kJ/mol)	ΔS^o^ J/(mol K)
295.15	−10.50	28.89	154.35	−7.14	24.61	138.32
303.15	−18.83			−12.12		
313.15	−27.15			−17.03		

* standard deviation (SD) = 2%.

**Table 4 nanomaterials-13-02572-t004:** Kinetic parameters for nickel adsorption on FEZ and EZ adsorbents *.

Adsorbent Material	Q_e_^exp^ (mg/g)	Pseudo-First Order			Pseudo-Second Order			Intraparticle Diffusion		
		Q_e_^calc^	K_1_	R^2^	Q_e_^calc^	K_2_	R^2^	K_i_	C	R^2^
FEZ	321.1	322.28	0.023	0.9994	321.58	2.781	0.9997	8.0276	33.875	0.9764
EZ	287.9	288.35	0.011	0.9991	287.95	1.924	0.9993	6.2782	21.466	0.9726

* standard deviation (SD) = 2%.

**Table 5 nanomaterials-13-02572-t005:** Comparison between nickel removal efficiency of adsorbents EZ and FEZ and those reported in the literature (selected studies) for materials similar to EZ and FEZ components.

Adsorbent Type	Removal Efficiency (%)	Ni Initial Concentration (%)	Reference
eggshell	90.9	100 mg/L	[69]
eggshell	93.5	100 ppm	[70]
eggshell-derived hydroxyapatite	91.0	100 mg/L	[49]
vinegar-treated eggshell waste biomass	76.5	1000 ppm	[71]
clinoptilolite	93.6	100 mg/L	[15]
zeolite	58.6	1000 mg/L	[40]
supported zeolite-Y hollow fiber membranes	63.0	10 mg/L	[72]
clinoptilolite	60.0	25 mg/L	[73]
EZ	97.3	25.5 mg/L	This study
FEZ	99.9	25.5 mg/L	This study

## Data Availability

The datasets and relevant material used and/or analyzed during the current study are available from the corresponding author on reasonable request.

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
