# Peer review of "Innovative Low-Cost Composite Nanoadsorbents Based on Eggshell Waste for Nickel Removal from Aqueous Media"

_nanomaterials, 2023, doi:10.3390/nano13182572_

Round 1
Reviewer 1 Report
In this paper, two new low cost nanoadsorbents (EZ and FEZ) were prepared for the retrieval of nickel from aqueous solutions. The composite nanoadsorbents remove nickel from aqueous solutions with high performance and adsorption capacities 321.1 mg/g and 287.9 mg/g, respectively. I recommend the acceptance of this paper after addressing the following minor points:
1) Please give some more discussion about the role of FeOOH in the retrieval of nickel from aqueous solutions.
2) What is the chemical force between the eggshell with zeolite/or FeOOH?
3) Calcium carbonate is the main component of egg shells. What will be happened if the eggshell is replaced by calcium carbonate?
4) There are some language errors in the text, and the paper should be corrected carefully and comprehensively.
In this paper, two new low cost nanoadsorbents (EZ and FEZ) were prepared for the retrieval of nickel from aqueous solutions. The composite nanoadsorbents remove nickel from aqueous solutions with high performance and adsorption capacities 321.1 mg/g and 287.9 mg/g, respectively. I recommend the acceptance of this paper after addressing the following minor points:
1) Please give some more discussion about the role of FeOOH in the retrieval of nickel from aqueous solutions.
2) What is the chemical force between the eggshell with zeolite/or FeOOH?
3) Calcium carbonate is the main component of egg shells. What will be happened if the eggshell is replaced by calcium carbonate?
4) There are some language errors in the text, and the paper should be corrected carefully and comprehensively.
Author Response
Reviewer response to comments are in the attached file

Reviewer 2 Report
Water pollution is a serious and global environmental problem for the health of human, the treatment of polluted water to clean water is urgently in nowadays. This manuscript reports two nanoadsorbents to remove nickel ions from aqueous solutions, which provides new results and meaningful conclusions to clean nickel ions pollution in water. Through detail analysis of chemical and physical properties and the measurement of adsorption capabilities, the nanosorbents of EZ and FEZ were demonstrated having high adsorption capabilities to remove nickel ions in aqueous solutions. The manuscript is meaningful and could be considered for publication after addressing the following comments.
1. In 2.1, the authors claimed that "Eggshells (ES) were collected from housework and washed four times with ultrapure water to remove any impurities." May this kind of ES too special to be reproduced by pioneers? It is better to give some more details about the ES. Besides, how can be authors confirm they removed impurities? What kinds of impurities ?
2. The BET surface area of eggshell is too small, how can the author accurately measure the area with the standard deviation as low as 0.02%?
3. The pore size distributions of the samples need to be displayed for the analysis of pore size differences.
4. The BET surface areas of EZ and FEZ were significantly higher when compared with eggshell ad zeolite, please provide a reasonable explanation.
5. The font was bold when discussing XRD, please revise.
6. The XRD pattern of EZ was missing in Figure 5.
7. In Table 5, the nickel removal efficiency should dependent on the concentration of initial nickel species and the amount of adsorbents, as well as the adsorption temperatures. Please also added these information to have a clearer comparison.
8. It is better to distinguish the roles of composites in EZ and FEZ in adsorbing nickel ions.
Should be improved for better reading and understanding.
Author Response
Reviewer comment responses are in the attached file

Reviewer 3 Report
The manuscript « Innovative low-cost composite nanoadsorbents based on egg-shell waste for nickel removal from aqueous media” aimed in the synthesis, characterization, functionalization of the eggshell base with the zeolite nanoparticles. The obtained novel materials are evaluate in Ni removal from aqueous solution.
The paper presents some interesting results and it can be useful for the readers of this journal. I recommend, major mandatory revision, before possible acceptation:
1) The abstract is too long and should be shortened, showing only the originality and interesting results of this work.
2) I see that in the section “ 2.1.Materials” authors describe the synthesis method of zeolite ? Zeolite preparation should be moved to the section 3 (3. Adsorbents preparation)
3) The BET analysis: scientific interpretation of the increases on specific surface area could be added.
4) XRD characterization: Authors could describe the method that used for particle size estimation. The particle size should be compared and confirmed with results obtained from SEM analysis.
5) All figures should be standardized and their qualities should be improved.
6) The results on Ni removal could be compared with other materials published elsewhere.
7) Authors sould add mechanism on the Ni adsorption over FEZ.
8) The number of figures should be reduced, also the number of tables. Authors can move figures and table in supporting file.
9) Conclusion is too long and should be shortened.
Author Response

(The authors gave the same response as above.)

Round 2
Reviewer 3 Report
Some revisions are needed before publication:
1) As reviewer the immobilization of FeOOH over eggshell/zeolite is not clear and authors could discuss the chemistry that used on this preparation (wich kind of interaction etc…).
2) Figure 4.1 ? please revise it.
3) 17 figures in a research article is a high number and I think that reducing the number could improve the quality of the article. In addition, I think that it was hard and difficult to read a long article with numerous figures and tables. When I request to reduce the number of figures, I am suggesting if authors’ tray to keep just the interesting figures and move the other figures in the supplementary file. If the authors prefer to keep all figures, I suggest bring together the figures of characterization in one figure, or the figures of Ni adsorption in one figure etc…. Of course, the qualities of the figures should be improved and standardized with the same size. Another suggestion: figures corresponding to the models are not necessary and it was well-known, but keeping just one table summarizing all datas will be better for the readers.
4) The mechanism of Ni adsorption over prepared materials should be explained. We need a scheme and reaction that involved removing Ni. It is the most interesting results that authors could added with details.
5) The authors said that conclusion in not long and can be accepted as is. I agree with the authors response but, again, reducing the number of word by keeping just the most interesting results could be very useful for readers.
Author Response
The response to reviewer comments are attached

Round 3
Reviewer 3 Report
Now the revised manuscript can be accepted as is.